# Learning Arborescence with An Efficient Inference Algorithm

## Abstract

We consider a class of structured learning problems on arborescence (i.e., the directed spanning tree) from the input graph. The key step involved in this problem is predicting the minimal weight arborescence (MWA) from the learned model. In literature, there are two lines of research for predicting MWA: the Chu-Liu Edmonds (CLE) (Chu & Liu, 1965) and the Lovász (Lovász, 1985) methods. The CLE method is easy to implement while it takes $\mathcal{O}(n)$ cycle contractions. Here $n$ is the graph size. The Lovász method reduces to the multi-pair shortest path (MPSP) problem and takes only $\mathcal{O}(\log n)$ contractions. Nevertheless, in the CPU setting, MPSP has the same time complexity as finding MWA. The Lovász method only attains time efficiency under a sufficient GPU setting. Both the aforementioned methods are painfully slow for large-scale learning tasks. In this research, we find the general MPSP problem can be simplified when working with machine learning models. This is because the learning model predicts edge weights for all pairs of vertices and the graph we process is always complete. Therefore, we only need to handle those paths that directly enter every weakly connected component (WCC) while the classic Lovász method need to handle all possible paths. This allows us to propose **La**zy Lo**vá**z (Lavá) method that enjoys $\mathcal{O}(\log n)$ contractions as well as efficient performance in both CPU and GPU settings. In experiments, we consider synthetic datasets and two real-world learning tasks, i.e., graph-based dependency parsing and unsupervised parsing on ListOps. The empirical results exhibit important gains of our Lavá method to the classic CLE and Lovász methods, that Lavá boosts the training time for arborescence learning tasks.

## 1 Introduction

This paper primarily focuses on a structured learning with the output structure being arborescence (i.e., directed spanning tree). Examples of real-world arborescence learning problems include graph-based dependency parsing (Koo et al., 2007) and ListOps parsing (Nangia & Bowman, 2018). At every step of learning, the neural model needs to infer the minimum weight arborescence (MWA). This inference procedure is mainly resolved by the Chu-Liu Edmonds algorithm (CLE) (Chu & Liu, 1965; Edmonds, 1967) or the Lovász method (Lovász, 1985).

The CLE method is straightforward in its implementation (Gabow et al., 1986; Mendelson et al., 2004). However, when dealing with larger-scale input graphs, CLE spends the majority of the time on cycle contractions. In fact, we show the CLE takes $\mathcal{O}(n)$ rounds of contractions theoretically and empirically. Here $n$ is the size of vertices in the graph. Lovász (1985) transformed the original finding MWA problem into the multi-pair shortest path (MPSP) problem. The Lovász method only needs $\mathcal{O}(\log n)$ contractions. Under sufficient GPU setting where the shortest path can be computed efficiently, Lovász algorithm attains faster running time over the CLE method. Later works marginally improve the classic Lovász with less GPU resource (Amato, 1993) and generalize to distributed network (Fischer & Oshman, 2021).

Since the learning model predicts the edge weights for all pairs of vertices, the graph we process is always complete. After the edge pre-process step, all the edge weights are non-negative (in Remark 1) and the graph is partitioned into several weakly connected components (WCCs). The classic Lovász method asks for computing the shortest path for every outside vertex $x_i$ into the cycle in the current WCC, which may travel around all the vertices and *go inside and then outside*

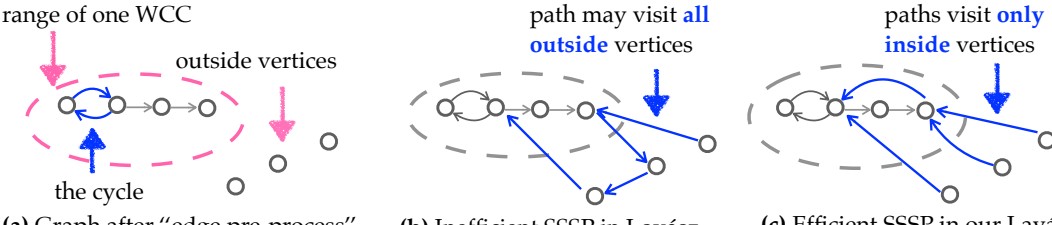

**(a)** Graph after "edge pre-process"    **(b)** Inefficient SSSP in Lovász    **(c)** Efficient SSSP in our Lavá

Figure 1: Our Lavá method is more efficient than the classic Lovász when computing the shortest path from outside to the cycle. **(a)** The pre-processed graph with circle denotes vertices and arrow ($\rightarrow$) denotes edges. "WCC" stands for weakly connected component. **(b)** In Lovász method, the outside vertex may find a path that wanders around over several WCCs, that involves all the vertices in the graph. **(c)** Our Lavá method computes those paths that directly enter the WCC and never go outside. Our method limits the scope of the shortest path and thus attains time efficiency.

of the current WCC. We observe that this can be simplified by considering those paths that directly enter the current component and *never go outside*. Such paths always exist because the graph we process is densely connected. We thus save the computation workload by limiting the scope of the shortest path. This observation allows us to attain time efficiency and thus boosts the training time for large-scale learning problems. Fig. 1 illustrates this idea with one potential graph.

In this research, we introduce, **La**zy Lo**vá**z (Lavá), an unified algorithm for finding MWA. Except for the above observation, we also introduce a bag of tricks for finding MWA on the large-scale and densely-connected graph. To detect the cycle and compute the shortest path, we exploit the property of matrix power to replace DFS-based algorithms. The whole method is described in the language of matrix computation. Therefore, our method is not only convenient and efficient for implementation with Numpy or Pytorch but also transparent to the CPU or GPU devices.

In experiments, we show the advantage of our Lavá approach in synthesized graphs and two real-world applications. In the synthetic experiment, we first show the empirical number of contractions by the CLE algorithm increases linearly with the graph size while our method takes much fewer contractions. Then we show that our Lavá method takes much less empirical running time to find MWA than CLE approach on a large-scale setting. Furthermore, for the dependency parsing task datasets and unsupervised parsing over Listops task, we show the proposed Lavá method attain show better running time than the CLE method over real-world datasets. Our contributions are:

**(1)** We propose an efficient inference algorithm (i.e., Lavá) for finding MWA, which is more compatible with and efficient for large-scale arborescence learning problems. **(2)** Theoretically, we show that our Lavá method takes $\mathcal{O}(\log n)$ and CLE takes $\mathcal{O}(\log n)$ round of contractions. We further show the correctness of our Lavá after adopting our observation on the shortest path. **(3)** We evaluate our method on synthetic data and two real-world applications. We show that Lavá is faster than CLE over all the datasets and tasks.

## 2 PRELIMINARIES

### 2.1 LEARNING ARBORESCENCE

**Notations.** Denote $G(V, E)$ as a weighted and directed graph with a given root $r \in V$. Root $r$ has no incoming edges and the rest vertices are strongly connected. The adjacency matrix $A$ that represents the connectivity of the graph is: for $x_i, x_j \in V$,

$$A_{i,j} = A(x_i, x_j) = \begin{cases} \phi(x_i, x_j) & \text{If } e(x_i, x_j) \in E \\ \infty & \text{If } e(x_i, x_j) \notin E \end{cases} \tag{1}$$

where $\phi(x_i, x_j) : V \times V \rightarrow \mathbb{R}$ denotes the weight of edge $(x_i, x_j)$ in the graph. We assume edge weights are all finite. All the incoming edges of vertex $x_i$ correspond to $i$-th column vector $A_{:,i}$ and all the outgoing edges can be found in $i$-th row vector $A_{i,:}$.

**Min-plus Product.** The min-plus product, is defined on the adjacency matrix for computing the shortest path in the graph (Williams & Xu, 2020). We denote "$\star$" as the min-plus product. Let

matrix $S$ contain all-pair *two-step* shortest distances, with $S_{ij}$ be the shortest distance of two-step path $x_i \rightarrow x_k \rightarrow x_j$. $S$ and $S_{ij}$ are computed as:

$$S = A \star A, \qquad S_{ij} = \min_{1 \le k \le |V|} A_{ik} + A_{kj}, \qquad (2)$$

The min-plus product can be easily implemented as generalized matrix-matrix multiplication using current GPU-based software, like Pytorch and JAX. The high parallelism of GPU greatly boosts the empirical running time for the min-plus product.

**r-arborescence.** The $r$-arborescence is a directed spanning tree with root $r$ (Korte & Vygen, 2018). It is a subgraph $T$ that covers all the vertices in the graph $G$ and there is a unique directed path in $T$ from $r$ to every other vertex $x_i \in V$. The weight of an arborescence $T$ is the summation of its edge weights (Jaini et al., 2018): $\phi(T) = \sum_{(x_i, x_j) \in T} \phi(x_i, x_j)$. The Minimum Weight $r$-Arborescence (MWA) is an $r$-arborescence whose $\phi(T)$ is minimum among all possible arborescences.

**Learning r-Arborescence.** The edge weight function $\phi_\theta$ is actually a neural network parameterized by $\theta$. Given a set of vertices $x = \{x_i\}_{i=1}^n$, the network predicts the possible edge weight $\phi_\theta(x_i, x_j)$ for every pair of vertices $x_i, x_j \in x$. The arborescence is represented as the binary matrix $T \in \{0, 1\}^{n \times n}$, where $T_{ij} = 1$ denotes edge $e(x_i, x_j)$ is included. Similarly, the score of arborescence is given by $\phi_\theta(T|x) = \sum_{(x_i, x_j) \in T} \phi_\theta(x_i, x_j)$. The probability of arborescence is defined as:

$$P_\theta(T|x) = \frac{1}{Z_\theta(x)} \exp(\phi_\theta(T|x)), \qquad \text{where } Z_\theta(x) = \sum_{T' \in \mathcal{T}(x)} \exp(\phi_\theta(T'|x)) \qquad (3)$$

$Z_\theta$ is the *partition function* and $\mathcal{T}(x)$ denotes all possible arborescences. In *inference*, given the trained parameters $\theta$ for the network, we predict the arborescence with the maximum score:

$$\arg \max_{T' \in \mathcal{T}(x)} \phi_\theta(T|x) = \arg \min_{T' \in \mathcal{T}(x)} \sum_{(x_i, x_j) \in T} -\phi_\theta(x_i, x_j) \qquad (4)$$

We can construct the adjacency matrix $A_\theta$ (in Eq. 1) using negative weight $-\phi_\theta(x_i, x_j)$ for all pairs of edge $e(x_i, x_j)$. It is exactly finding the MWA for $A_\theta$. We brief the inference procedure in Sec. 2.2.

Given a training set $\mathcal{D} = \{x^k, T^k\}_{k=1}^N$, where each $T^k$ is human-labeled output for input $x^k$. Learning can be achieved via maximal likelihood estimation. In other words, we find the optimal parameters $\theta^*$ by minimizing the negative log-likelihood: $\ell_\theta(\mathcal{D}) = -\sum_{k=1}^N \phi_\theta(T^k|x^k) + \log Z_\theta(x^k)$. Recently, Paulus et al. (2020); Struminsky et al. (2021) propose the recursive Gumbel-max trick, that we can estimate the term $\log Z_\theta(x)$ using the Gumbel distribution:

$$\log Z_\theta(x) = \mathbb{E}_{g'} \left( \max_{T' \in \mathcal{T}(x)} \sum_{(x_i, x_j) \in T} \phi_\theta(x_i, x_j) + g'_{ij} \right), \qquad g'_{ij} \sim \text{Gumbel}(0, 1) \qquad (5)$$

We can use Eq. 4 to predict the set of edges on MWA with the *perturbed* adjacency matrix $A_\theta + g'$ and compute the corresponding arborescence score. The parameters $\theta$ can be trained using gradient descent: $\theta^{t+1} = \theta^t - \eta \nabla \ell_\theta(\mathcal{D})$, where $\eta$ is the learning rate and $\nabla \ell_\theta(\mathcal{D})$ is the gradient of the negative log-likelihood function.

## 2.2 FINDING MIN-WEIGHT ARBORESCENCE

Chu-Liu Edmonds (CLE) (Chu & Liu, 1965; Tarjan, 1977) is the most well-known method for finding MWA. Later, Lovász (1985); Amato (1993); Fischer & Oshman (2021) proposed a theoretically efficient variant by optimizing over the worst case in CLE. We brief CLE and Lovász methods below.

**Chu-Liu Edmonds.** 1) For every non-root vertex $x_i$ ($x_i \ne r$), find the incoming edge with minimum weight then subtract this minimum value from all the incoming edges of vertex $x_i$. 2) Next extract all the vertices along with all the edges with weight being zero. They form a new graph $F$. If there are *multiple zero-weight edges*, then randomly pick one incoming edge for every vertex. If $F$ has no cycle, it is clearly the minimum one. So the contraction terminates. 3) Otherwise, there exists at least one cycle. Every cycle is then contracted into a single *super vertex*. Those self-connected edges are removed. The resulting graph is noted as $G'$.

The above three steps are recursively applied over graph $G'$ until the final contracted graph is an arborescence. On termination of the contraction, the expansion procedure is launched to recursively expand those super vertices (i.e., contracted cycles) and determine which edge on the cycle should be discarded. Finally, it outputs the set of edges of MWA in the original graph. Figure 2(a) sketches the connection of our Lavá and CLE methods.

**Lovász's Meta Algorithm.** Lovász (1985) considered the worst case of CLE, noted as *nested cycle contractions*. The CLE method might spend many steps contracting nested cycles. The Lovász approach reduces to the multi-pair shortest path (MPSP) problem and only takes $\mathcal{O}(\log n)$ contractions in the worst case. Specifically, the Lovász approach finds the *shortest entering path* into a cycle. The algorithm not only contracts those vertices on the cycle but also those vertices relevant to the shortest entering path. The whole contraction process is detailed in Appendix A.1. For those follow-up works, Amato (1993) reducing the dependency on processors in the GPU setting and Fischer & Oshman (2021) improved the running time in distributed CONGEST network. Figure 1(b,c) illustrates the major difference between our Lavá and Lovász methods.

## 3 METHODOLOGY

For the task of learning arborescence from training data, the neural network $\phi_\theta$ takes in all vertices $x_1, \ldots, x_n$ as input and outputs a set of edges $T$ that form the arborescence with maximum score (as in Eq. 4) at every step of gradient computation. Specifically, the network predicts the edge score for every pair of vertices $\phi_\theta(x_i, x_j)$ and constructs the adjacency matrix $A_\theta$. As mentioned in Sec. 2.2, CLE method takes this matrix $A_\theta$ as input and outputs the set of edges that forms MWA by recursive contractions and expansions. The CLE method is insufficient because it spends most of the time shrinking to smaller and smaller graphs and later expanding to larger and larger graphs. In fact, we show in Sec. 3.2 that CLE requires $\mathcal{O}(n)$ contractions and expansions, under the mild assumption of the graph. Since large-scale inputs are the norm in real-world settings, this limits the scalability of those machine learning tasks that relies on the efficient inference of MWA.

As mentioned in Sec. 2.2, Lovász (1985) solved the nested cycle contractions problem, that reduces from $\mathcal{O}(n)$ contractions to $\mathcal{O}(\log n)$ contractions. Nevertheless, Lovász's meta-algorithm itself (Lovász, 1985) and its variants (Amato, 1993; Fischer & Oshman, 2021) need to compute the multi-pair shortest path (MPSP) problem to find the *shortest entering path* into the cycle. Specifically, every outside vertex needs to find the shortest path to the cycle, where the path could come inside the WCC that contains the cycle and go outside. In the worst case, the path may touch all the outside vertices. Solving the intermediate MPSP problem makes Lovász time-consuming.

In this research, we find the general MPSP problem can be simplified when working with neural networks. Since the graph constructed by the neural network is always complete, a path once enters the WCC should never go outside. By ruling out those paths that wander around several WCCs, we only consider those paths that start from one outside vertex and directly enter the WCC. Our Lavá methods solve a much smaller scale problem and thus gain computational efficiency compared with the Lovász method. This allows us to propose **La**zy Lo**váz** (Lavá) method that enjoy $\mathcal{O}(\log n)$ contractions with friendly and efficient implementation in CPU and GPU settings. For example, Figure 1(b,c) presents our Lavá method can better handle the shortest entering path problem than Lovász method. Figure 1(a) presents our Lavá takes fewer contractions than CLE method.

### 3.1 FINDING MIN-WEIGHT ARBORESCENCE WITH LAZY LOVÁSZ ALGORITHM

Given the adjacency matrix $A \in \mathbb{R}^{n \times n}$ (in Eq. 1), the task is to output those edges that form the MWA. Assume root $r$ has the largest index and the MWA is unique. Also given the neural network $\phi_\theta$, there exists an edge for every pair of vertices $x_i, x_j$ ($x_j \neq r$) with weight $\phi_\theta(x_i, x_j)$. We describe the whole contraction process in the language of matrix computation. The expansion step is omitted here since it is the same as all the classic methods.

**Step 1: Edge Pre-process.** This step finds the minimum incoming edge for each vertex. For vertex $x_j$ ($1 \leq j < n$), we go through the $j$-th column vectors of the matrix $A$. Denote $\pi_j$ as the index for the minimum incoming edge for vertex $x_j$:

$$\pi_j = \arg \min_{1 \leq i \leq n} A_{i,j}, \quad \text{for } 1 \leq j < n \tag{6}$$

where $e(x_{\pi_j}, x_j)$ is the corresponding edge for vertex $x_j$. Then we need to subtract the value of the minimum incoming edge from all the incoming edges of every vertex. This is computed as the $j$-th column vector $A_{:,j}$ subtracting $A_{\pi_j,j}$:

$$A_{i,j} = A_{i,j} - A_{\pi_j,j}, \qquad \text{for } 1 \le i < n \tag{7}$$

We denote the resulting graph with edges determined by $\{e(x_{\pi_j}, x_j)\}_{j=1}^{n-1}$ as $F$. We also use $F$ to denote the corresponding adjacency matrix $F \in \{0,1\}^{n \times n}$ with $F(x_{\pi_j}, x_j) = 1$ for $1 \le j < n$.

**Remark 1.** *All the incoming and outgoing edge weight become non-negative after the first "Edge Pre-process" operation: $A(x_i, x_j) = A(x_i, x_j) - \min_{x_k} A(x_k, x_j) \ge 0$, for all $x_i, x_j \in V$.*

**Step 2: Termination Criterion.** The algorithm terminates when graph $F$ is acyclic. We decide if $F$ is acyclic by counting the number of weakly connected components (WCCs) in $F$. Specifically, a graph is said to be weakly connected if replacing all of its directed edges with undirected edges produces a connected (undirected) graph. One WCC implies $F$ is connected and acyclic so that $F$ is an arborescence. Otherwise, graph $F$ contains at least one cycle.

Specifically, we compute binary matrix power series to get WCCs, since binary computation is more efficient than arithmetic operators, like decimal multiplication. Consider $k$-th matrix power:

$$F^k = \underbrace{F \times F \times \cdots \times F}_{k \text{ times multiplications}}, \qquad \text{where } (F \times F)_{ij} = \bigvee_{1 \le k \le n} (F_{ik} \wedge F_{kj}) \text{ for } 1 \le i, j \le n \tag{8}$$

Operator "$\times$" denotes binary matrix multiplication. "$\vee$" and "$\wedge$" means element-wise logical-OR and logical-AND operators correspondingly. $F_{ij}^k = 1$ if and only if vertex $x_i$ can reach vertex $x_j$ with *exactly* $k$ steps using edges in graph $F$. The reachability matrix for all pairs of vertices *within* $n$ steps is:

$$S = \bigvee_{k=1}^{n} F^k = F \vee F^2 \vee \ldots \vee F^n = (I \vee F)^{n-1} \times F \tag{9}$$

We need $\mathcal{O}(\log n)$ iterations to compute the matrix $(I \vee F)^{n-1}$ using the idea of divide and conquer. See detailed illustration in Appendix C.2. We find $S_{ii} = 1$ implies vertex $x_i$ is on the cycle, as there is a path in $F$ that vertex $x_i$ can reach itself. Thus, the termination criterion is $\sum_{i=1}^{n} S_{ii} = 0$.

This is not the only way to determine graph connectivity, prior works invoke matrix inverse or DFS to detect the connectivity of graph $F$. We give an in-depth discussion in Appendix C.3. Matrix inverse cannot scale to a large matrix due to numerical precision. DFS is efficient in serial (i.e., CPU) setting but is undesirable under parallel (i.e., GPU) settings. Therefore, we consider this simple matrix computation-based alternative to find WCCs and determine termination.

**WCCs Extraction.** This is an intermediate step for the following shortest entering path. We need to partition the vertices $V$ using matrix $S$ (in Eq. 9). For $i$-th WCC, we extract 1) $C_i$, the set of vertices on the cycle. If $x_i$ is on the cycle, we can locate the rest using the $i$-th column and row vector, i.e., $S_{ij} = 1 \wedge S_{ji} = 1$. 2) $H_i$, the set of vertices that can be reached from the cycle but not on the cycle, i.e., $C_i \cap H_i = \emptyset$. 3) $K_i$, the set of vertices that are non-reachable from the cycle.

$$\begin{aligned} C_i &= \{x_j | S_{ij} = 1 \wedge S_{ji} = 1, \text{ for } 1 \le j < n\}, \\ H_i &= \{x_j | S_{ij} = 0 \wedge S_{ji} = 1, \text{ for } 1 \le j < n\}, \\ K_i &= \{x_j | S_{ij} = 0 \wedge S_{ji} = 0, \text{ for } 1 \le j \le n\}. \end{aligned} \tag{10}$$

We obtain a list of vertices $\{(C_i, H_i, K_i)\}_{i=1}^{L}$ that partition the graph. See Example 1 for illustration.

**Step 3: Lazy Shortest Entering Path.** Fix $i$-th WCC, we compute the shortest entering path from vertices in $K_i$ to vertices in $C_i$ via edge relaxation. Briefly speaking, we consider $x_k \to x_c$ for every $x_k \in K_i, x_c \in C_i$, that corresponds to the one-step path. Then we try to relax the distance with $x_k \to x_h \to x_c$ for $x_h \in H_i$, that are those two-step paths. The above edge relaxation step terminates once the distance cannot be relaxed anymore.

Specifically, denote adjacency matrices $A_{K_i, C_i}$ to represent the connectivity between the set of vertices in $K_i$ and $C_i$. Matrices $A_{H_i, C_i}$ and $A_{H_i, H_i}$ are defined similarly. To avoid recursive edge relaxation, we use the same matrix power trick to compute the all-pair shortest distance for $x_h, x_h' \in H_i$. Let matrix $D$ be the resulting matrix, we have: $D = A_{H_i, H_i} \star A_{H_i, H_i} \star \cdots \star A_{H_i, H_i}$ with at

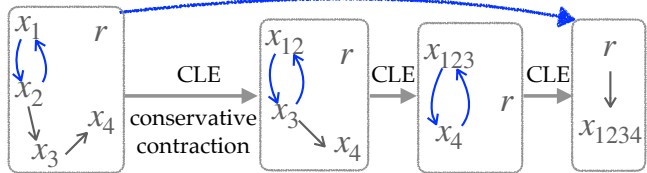
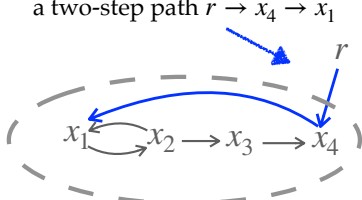

**(a)** contraction in our Lavá are more progressive than in CLE      **(b)** example of entering path in Lavá

Figure 2: **(a)** The Lavá method contracts vertices more progressively than the CLE method. **(b)** The shortest entering path picks the minimum one (in Eq. 11). The outside vertex $K = \{r\}$ can either directly enter cycle $C = \{x_1, x_2\}$ or by a intermediate vertex in $H = \{x_3, x_4\}$. Lavá method find the shortest entering path $(r \rightarrow x_4 \rightarrow x_1)$ and directly contracts $\{x_1, x_2, x_3, x_4\}$ as $x_{1234}$.

most $\log |H_i|$ min-plus products. In comparison, the classic Lovász method would use the whole matrix $A$ instead of $A_{H_i}$, computing $D = A \star \ldots \star A$ with at most $\log n$ min-plus products. The min-plus product $\star$ is defined in Eq. 2.

Let $\beta_i \in \mathbb{R}$ be the shortest entering distance from $K_i$ to $C_i$. $\beta_i$ could be picked from: 1) one-step paths, $x_k \rightarrow x_c$, $x_k \in K_i, x_c \in C_i$. 2) two-step paths, $x_k \rightarrow x_h \rightarrow x_c$, $x_h \in H_i$. 3) more than two-step paths, $x_k \rightarrow x_h \rightarrow x'_h \rightarrow x_c$. Note that the intermediate vertices are only picked from $H_i$, $x_h, x'_h \in H_i$. It can be summarized as:

$$\beta_i = \min \begin{cases} A_{K_i,C_i} & \text{Case 1: one-step paths} \\ A_{K_i,H_i} \star A_{H_i,C_i} & \text{Case 2: two-step paths} \\ A_{K_i,H_i} \star D \star A_{H_i,C_i} & \text{Case 3: more than two-step paths} \end{cases} \tag{11}$$

Example 1 and Fig. 5(b) illustrate this procedure.

**Step 4: Contraction.** Given $\beta_i$, we contract vertices on cycle $C_i$ and those relevant vertices in $H_i$. Denote the set of vertices to be contracted as $U_i$,

$$U_i = C_i \cup \{x_h | \min_{x_c \in C_i} A(x_h, x_c) \leq \beta_i, x_h \in H_i\} \tag{12}$$

Contraction is done by picking an arbitrary vertex in $u \in U_i$ as the representative and using a mask vector to filter out the rest vertices in $U_i$. The weight of incoming edges of vertex $u_i$ in the contracted graph $G'$ would be updated as follow:

$$A(x', u_i) = \min_{x_u \in U_i, x_c \in C_i} A(x', x_u) + A(x_u, x_c) - \beta_i \tag{13}$$

All the remaining edges stay the same. Afterward, we iteratively apply the above 4 steps over graph $G'$ until there are no cycles. We use Example 1 to show the detailed steps of our Lavá method.

**Example 1.** Take Fig. 2 as an example. In Fig. 2(a), Lavá contracts are more progressive than the classic CLE method. The vector $\pi$ (in equation 6) for the most left figure is: $\pi = [2 \quad 1 \quad 2 \quad 3 \quad -1]$, where $\pi_3 = 2$ means the minimum incoming edge of vertex $x_2$ is $e(x_3, x_2)$. "$-1$" implies root $r$ has no incoming edge. Also this most left graph has two WCCs: $\{r\}$ and $\{x_1, x_2, x_3, x_4\}$. Figure 2(b) computes the shortest entering path. According to Eq. 10, we have $C = \{x_1, x_2\}$, $H = \{x_3, x_4\}$ and $K = \{r\}$. We consider two possible cases for this example: 1) those one-step paths (in Case 1), which are recorded in matrix $A_{K,C}$. 2) those two steps paths (in Case 2), like $r \rightarrow x_3 \rightarrow x_1$, which can be computed as $A_{K,H} \star A_{H,C}$,

$$A_{K,C} = [\phi(r, x_1) \quad \phi(r, x_2)], \qquad A_{K,H} \star A_{H,C} = [\phi(r, x_3) \quad \phi(r, x_4)] \star \begin{bmatrix} \phi(x_3, x_1) & \phi(x_3, x_2) \\ \phi(x_4, x_1) & \phi(x_4, x_2) \end{bmatrix}$$

The min-plus operator $\star$ can be: $(A_{K,H} \star A_{H,C})_{1,1} = \min_{x' \in H}\{\phi(r, x') + \phi(x', x_1)\}$. The shortest entering path is $r \rightarrow x_4 \rightarrow x_1$, then the set of vertices to be contracted is $\{x_1, x_2, x_3, x_4\}$.

Our proposed method described in matrix computations can be easily implemented using Numpy on CPU or Pytorch on GPU. The matrix computation offers a transparent protocol to compute on GPU in parallel, without exactly manipulating every parallel processor on GPU devices. Furthermore, the matrix-based approach allows us to conveniently process the output from the neural network. The method may also be extended to parallel computing but would require remodeling each step, we leave this discussion in Appendix B.

## 3.2 THEORETICAL INSIGHTS ON LAVÁ METHOD

**Analysis on Rounds of Contractions.** The main impacting factor for the running time of the proposed method is the number of contractions. Let $T_n$ be the number of contractions for the graph with $n$ vertices. In the CLE method, we consider that the graph can be contracted into smaller graphs of sizes $\{2, \ldots, n-2\}$ *with equal probability*. This simplified assumption may be different from the real-world setting but it can help us to reveal the averaged case of the CLE algorithm. We leave the study with more rigorous assumptions on graph contraction as future work. We have the following recursion: $T_n = 1 + \frac{1}{n-3} \sum_{t=2}^{n-2} T_t$. We obtain $T_n = \mathcal{O}(n)$ by solving this recursion. This implies, other than the rarely-happen worst case, CLE method on average takes $\mathcal{O}(n)$ contractions under the equal probability assumption. Furthermore, we show in Theorem 1 that Lavá takes at most $\mathcal{O}(\log n)$ contractions, which is more efficient than the CLE method.

**Theorem 1** (Rounds of Contractions). *In Lavá method, every directed cycle in $C_i$ contains **at least two** super vertices from the previous round. Lavá takes at most $\mathcal{O}(\log n)$ contractions.*

*Sketch of Proof.* We show there exists at least one super vertex in the cycle and we prove by contraction that there exist at least two super vertices. Finally, we do induction on the contraction rounds that give at most $\mathcal{O}(\log n)$ round of contractions for the whole procedure. Details in Appendix D. □

According to Lemma 1, by induction on the contraction rounds, we conclude that the Lavá method finds the MWA for graph $G$.

**Lemma 1** (Correctness). *$T^* = \arg\min_{T \in \mathcal{T}} T$ represents the MWA $T^*$ in the original graph $G$ with weight $\phi(T^*)$. and $T'^* = \arg\min_{T' \in \mathcal{T}'} T'$ represents the MWA $T'^*$ in the contracted graph $G'$ with weight $\phi(T'^*)$. Then we have: $\phi(T^*) = \phi(T'^*) + \sum_{i=1}^{L} \beta_i$.*

## 4 RELATED WORK

**Finding MWA.** Historically, Chu-Liu Edmonds (CLE) algorithms (Chu & Liu, 1965; Edmonds, 1967; Bock, 1971; Tarjan, 1977) along with its efficient data structures (Gabow et al., 1986; Mendelson et al., 2004) are proposed to find the min-weight arborescence in the directed graph. But those data structures, like a heap, are left to be developed for GPUs. Most recently, Böther et al. (2022) benchmark all the implementations. Meanwhile, Humblet (1983) first proposed a distributed computing scheme for CLE approach. Lovász (1985) proposed to solve the nested cycle contraction problem, but it asks for $\mathcal{O}(|V|^3)$ many processors for solving the intermediate shortest path problem in parallel. Later, Lucas & Sackrowitz (1992); Amato (1993) reduced to the dependency on processors. Fischer & Oshman (2021) adopted recent advances in distributed shortest path (Elkin, 2020) and extend the Lovász algorithm to distributed CONGEST network. Another line of work is based on constrained linear programming (LP) (Fischetti & Vigo, 1997; Király et al., 2020). However, LP needs exponential many constraints to enforce the output is an arborescence.

**Learning Arborescence.** Inference-based learning can be mainly divided into two categories: discriminate max-margin learning and probabilistic perturb-and-MAP learning. The first category, including structured SVM (Tsochantaridis et al., 2004) and max-margin Markov network (Taskar et al., 2003), learns to minimize the gap between the output in the dataset and the best-predicted output. The second category focuses on differentiable structured learning for arborescence. Based on the perturbed optimizer theory (Berthet et al., 2020), they utilize perturbed inference for estimating the partition function (Niculae et al., 2018; Paulus et al., 2020; Struminsky et al., 2021).
There also exists a diverse category of applications that involve arborescence. In graph-based dependency parsing, MWA denotes the best grammar tree of input sentences (McDonald et al., 2005; Ma & Hovy, 2017; Zhang et al., 2019). In causality learning, MWA represents the optimal causal additive trees (Jakobsen et al., 2022). In multiple human tracking, MWA denotes the best tracking association for the objects (Henschel et al., 2014). MWA is applied for finding the optimal vessel tree reconstruction (Zhang et al., 2021). MWA is adopted to decompose the Markov chain in game theory (Newton & Sandholm, 2021).

## 5  EXPERIMENTAL ANALYSIS

We mainly show Lavá method serves as an efficient alternative for CLE method in the arborescence learning applications and synthesised dataset. Furthermore, we show Lavá uses much less contractions on real-world and synthesis dataset and use less time to detecting WCCs, compute shortest entering distance when comparing to its baselines.

### 5.1  SYNTHETIC DENSE GRAPH

**Empirical Running Time.** The experiment settings are mentioned in Appendix G.2. Fig. 3(a) shows our Lavá takes much less number of contractions empirically. We observe that larger graphs requires more contraction steps than smaller graph in the CLE approach, which is time-consuming for the learning problem in large-scale data. We also notice Lavá and the classic Lovász takes same amount of contractions for the same input graph. We then compare the empirical running time of the proposed approach with existing baselines over synthetic datasets (in Fig. 3(b)). Then the same input is feed into all the competing methods and we measure the time of every algorithm need to get an optimal spanning arborescence. We repeat 100 times and use the averaged running time as the running comparison metric. As we can see from Fig. 3(b), when the size of graph is smaller than $2^9$, both of the methods attain comparable results. When graph size is larger than $2^{10}$, the proposed Lavá takes significant less time than the classic CLE method. Since CLE needs to build up and then reset all the necessary data structures, like Heap and disjoint set, when reading every input. While Lavá only needs to process the matrix and run the binary and min-plus matrix multiplication.

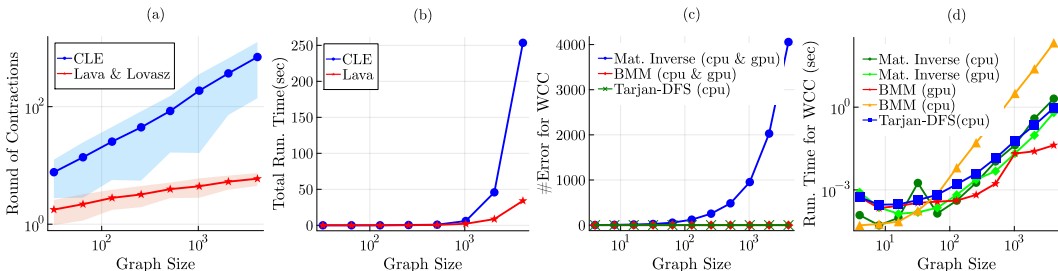

Figure 3: **(a)** The round of contractions taken by the CLE method grows linearly with graph size while our Lavá method takes much less rounds of contractions. **(b)** Empirical running time comparison between Lavá and CLE methods on synthetic datasets. Lavá takes significant less time to process the graph with vertices $n \geq 2^{10}$. **(c)** Our proposed BMM (in Eq. 9) and Tarjan-DFS computes the exact matrix $S$ and attain zero numerical error. Matrix inverse (Mat. Inverse) cannot be applied to detect WCCs as the number of mismatch in the computed matrix $S$ accumulates linearly with the graph size. **(d)** BMM executed on GPU takes much less time for detecting WCCs than all the competing approaches. Here, "BMM" and "Mat. Inverse" is abbreviation for binary matrix multiplication and matrix inverse correspondingly.

**WCCs Implementation Analysis.** Here we compare with one baseline (Kepner et al., 2016) that propose to use matrix inverse for detecting WCCs. The detailed process of matrix inverse for computing WCCs is in Appendix C.3. The error is defined as the number of mismatches between the computed matrix $S$. In Fig. 3(c), we observe the error for detecting WCCs made by the matrix inverse approach (Mat. Inverse) becomes obvious when the graph is large while our method computes WCCs with no errors (as in Eq. 9). That's one major reason that we use binary matrix multiplication (BMM) to detect WCCs. In Fig. 3(d), we observe that BMM deployed on GPU attain the fastest approach for detecting WCCs than Tarjan's DFS algrorithm as well as Matrix inverse methods.

### 5.2  GRAPH-BASED DEPENDENCY PARSING

Dependency parsing considers the syntactic structure of a sentence, which describes the grammatical relations among words (Jurafsky & Martin, 2009). Words in sentences are formulated as vertices a in graph, that $x = [x_1, \ldots, x_{n-1}, r]$ represents an input sentence with $t$ being the dummy root. Let $T$ denotes the ground-truth dependency tree for sentence $x$. The annotation tree $T$ is composed of $n - 1$ edges, where every $(v_{x_i}, v_{x_j}) \in T$ is a directed dependency relation from head word $x_i$

Table 1: Lavá takes less contractions and time compare with baselines on three dependency parsing datasets. Both of them predict the same perturbed MWA when estimating $\log Z$ (in Eq. 5). "ApxErr" stands for the approximation error from the estimated value by perturbed inference to the exact value of $\log Z_\theta$. "#contract." denotes the averaged number of contractions; "Infer." means the averaged time to inference the MWA by each algorithm.

| Methods | English - GWT | | | Dutch - Alpino | | | French - GSD | | |
|---|---|---|---|---|---|---|---|---|---|
| | ApxErr | #contract. | Infer. | ApxErr | #contract. | Inf. time | ApxErr | #contracts | Infer. |
| CLE | 0.14 | $13.6_{10.4}$ | $0.09_{0.12}$ | 0.18 | $17.1_{7.9}$ | $0.14_{0.20}$ | 0.22 | $25.2_{12.7}$ | $0.19_{0.21}$ |
| Lovász | 0.14 | $\mathbf{3.4_{1.2}}$ | $\mathbf{0.08_{0.08}}$ | 0.18 | $\mathbf{4.0_{0.7}}$ | $\mathbf{0.11_{0.21}}$ | 0.22 | $\mathbf{6.5_{0.7}}$ | $\mathbf{0.17_{0.13}}$ |
| Lavá (ours) | 0.14 | $\mathbf{3.4_{1.2}}$ | $0.15_{0.22}$ | 0.18 | $\mathbf{4.0_{0.7}}$ | $0.29_{0.34}$ | 0.22 | $\mathbf{6.5_{0.7}}$ | $0.43_{0.27}$ |

to modifier word $x_j$. The learning task is to learn a neural model given annotated training data $\{x_i, T_i\}_{i=1}^N$. In inference, we need to predict a dependency tree with optimal score for a given input $x'$. We leave the detailed experiment configurations in Appendix G.1.

**Comparison.** Using the same deep neural network, we ask every baselines to predict the MWA over three multilingual datasets for the dependency parsing task. The results are collected in Table 1. In terms of the number of contractions, our Lavá and the classic Lovász uses less contractions than the CLE method. In terms of the empirical running time for inference MWA, our Lavá attain much less time compare with Lovász because it handle much smaller SSSP problem while Lovász need to handle all-pair shortest path problem. Furthermore, our Lavá takes less time compare with the classic CLE approach. The Finally, we use the same perturbed adjacency matrix to estimate the $\log Z_\theta$ terms. task in described in Appendix G.1. All the three methods return the same correct MWA and attain the same approximation error.

## 5.3 UNSUPERVISED PARSING ON LISTOPS

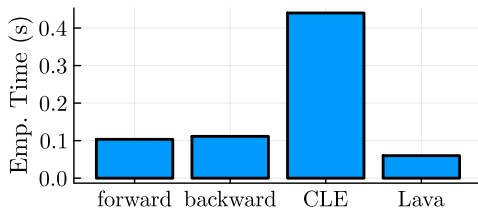

Figure 4: Empirical running time on ListOps task. Our Lavá method takes less time than CLE method, which can be a better alternative for the arborescence learning problem.

The ListOps task contains sequences of prefix arithmetic expressions, e.g., `[MAX 29 [MIN 47 ] 0 ]`. The arithmetic syntax for the input sequence induces a arborescence with the first token being the root. We follow the same graph neural network as described in Paulus et al. (2020). We benchmark the time of inferencing the MWA with the adjacency matrix induced by the deep network, the forward propagation of the neural network as well as the backward gradient pass for the neural network. The results are collected in Fig. 4, where we can see the classic CLE becomes one major time-consuming component while our Lavá can serves as an efficient alternative to boost the training speed of the network.

## 6 CONCLUSION

In this research, we investigate the inference problem when learning arborescence. We observe that the inference module (i.e., CLE method) that is invoked at every step of gradient computation becomes time-consuming on the large-scale dataset. Therefore, we propose Lazy Lovász for efficient inference of the optimal output of the learning model. Compared with CLE method, it needs $\mathcal{O}(\log n)$ rounds of contractions instead of $\mathcal{O}(n)$ by the CLE approach. Compared with the classic Lovász, it reduces the computational workload for the shortest entering path problem. In experiments, we show the empirical running time analysis on synthesized data. Furthermore, we conduct experiments on two real-world tasks: graph-based depdnency parsing and latent tree learning on ListOps. All the experiment results suggest that Lavá can efficiently find MWA for the network's output. This study implies for learning over large-scale arborescence learning problems, one could potentially give our Lazy Lovász a try.

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
