# OpenReview forum: "Learning Arborescence with An Efficient Inference Algorithm"
_ICLR.cc/2023/Conference — Submitted to ICLR 2023_

### Official Review · Reviewer_kqon · 2022-10-24

**Confidence:** 4
**Correctness:** 4
**Technical Novelty And Significance:** 2
**Empirical Novelty And Significance:** 2
**Recommendation:** 3

**Clarity, Quality, Novelty And Reproducibility:**

From my point of view, the paper is not very well organized, with notably a number of redundancies in the positioning in the literature.

The part dedicated to learning is insufficiently clear and detailed. For example, I did not understand the idea of learning the weight function $\phi$ of the graph since it is supposed to be known. The structure of the network and the way it is trained should be investigated in depth. The generation of the training data seems to me to be a difficult problem which, unless I am mistaken, is not addressed at all in the article.

**Strength And Weaknesses:**

The main strength of this paper is that it is rather complete: it introduces a new algorithm for a difficult problem and analyses it both theoretically and numerically. However, it suffers from two important flaws:
- First, the announced complexity in O(log n) contractions for the MPSP problem puts the proposed algorithm at the same level as the Lovasz algorithm developed in 1985. (The authors also announce that their algorithm benefits from "friendly and efficient" CPU and GPU implementation, but they should be more precise on this point. In particular, they should precisely compare their contribution and Lovasz algorithm on this aspect.)
- Second, the experimental analysis is mainly dedicated to the comparison with the Chu-Liu Edmonds method, introduced in 1965, which is theoretically worse because it needs O(n) contractions.
For these two reasons, the contribution is poorly situated with respect to the state of the art, both theoretically and numerically. The authors mention themselves more recent references than the two mentioned above, which should serve as baselines.

**Summary Of The Paper:**

This paper presents a new algorithm, referred to as the LAVA method, for evaluating the minimal weight arborescence of a directed graph which each of the edges carries a weight. It was shown in the literature that this question can be reduced to the Multi-Pair Shortest Path (MPSP) problem. The present paper investigates this problem through deep learning methods: it introduces a new algorithm based on neural networks, states a result about its time-complexity, and shows numerical results and comparisons with existing methods of the literature.

**Summary Of The Review:**

The proposed algorithm certainly deserves attention, but, in my opinion, both the presentation and the theoretical and numerical results do not allow the publication of this article in its current form.

---

> ### Author Response · Authors · 2022-11-07
> **Reply to reviwer kqon**
>
> Dear reviewer,
>
> Thank you for the valuable comments. We would like to work our best to address your concerns answer your questions.
>
> - Regarding “poor presentation”
>
> Thanks for pointing this out. We will carefully rewrite the content and make sure everything is proofread.
>
> - Regarding “complexity in O(log n) contractions”
>
> Our Lava method uses the same round of contractions compared to all the existing Lovasz-based approaches. We simplify the MPSP during every step of contraction because of two observations when working with the neural network’s outputs: 1) the graph is always complete. Since the neural model predicts the edge weight for every pair of vertices. and 2) the edge weight is non-negative after the first preprocessing step.
>
>
>
> - Regarding “baselines in experiments”
>
> We acknowledge that more baselines would make the comparison more solid. The distributed Lovasz method needs a cluster of 10 computers to process a graph of 10 vertices, which is impractical to compare within experiments. We hope our argument serves as a reasonable argument to this question.
>
> - Regarding “learning the weight function”
>
> Sorry for the confusion on this part. Here we use dependency parsing as an example. The input is a list of words in the sentence. The task requires predicting the best grammar tree for all the words in the sentence. The neural network would predict the edge weight between every pair of words. The inference step is to find an arborescence (directed spanning tree) for the input sentences. The function $\phi_{\theta}$ represent all those neural networks that take in sentences as input and outputs weight score for all pairs of words.

---

### Official Review · Reviewer_sBZy · 2022-10-24

**Confidence:** 2
**Correctness:** 3
**Technical Novelty And Significance:** 2
**Empirical Novelty And Significance:** 2
**Recommendation:** 3

**Clarity, Quality, Novelty And Reproducibility:**

For the organization, related work can be placed in the introduction part, and more explanations are needed to clarify the strength of the algorithm compare to the Lovasz algorithms.

**Strength And Weaknesses:**

Strength:
1. They cleverly use the observation that edge weights for all pairs of vertices and the graph always complete in learning model, they modify the prior algorithms by restricting the scope, which can decrease the computation burden.
2. Consider the algorithm in GPU setting, they change the traditional operation to GPU-friendly operation.

Weaknesses:
1. I am not sure that the contribution part "Theoretically, we show that our Lava method takes O(log n) and CLE takes O(log n) round of contractions." whether the CLE takes O(log n) is typo.
2. In numeric experiment, the  proposed algorithms seems not better than Lovasz algorithms, in some cases even worse than CLE. They claim that their algorithm is suitable for large-scale, but they actually do not test large-scale real dataset.
3. The idea of the paper needed to be explained more clearly.

**Summary Of The Paper:**

This research considers structured learning problems on arborescence from input graphs. Prior algorithms are CLE, Lovasz algorithms, and their variants. The authors find the general MPSP problem(sub-problem of the original problem) can be simplified when working with machine learning models. This is because the learning model predicts edge weights for all pairs of vertices and the graph is always complete. Through this observation, they can restrict the scope of the original problem and thus decrease the computation burden. Some other tricks are also mentioned for large-scale learning tasks.

**Summary Of The Review:**

This research considers structured learning problems on arborescence from input graphs. The authors find the general MPSP problem(sub-problem of the original problem) can be simplified when working with machine learning models, they mention that their algorithms are suitable for large-scale data, but none of theoretical analysis and numeric experiments are provided for their algorithms.

---

> ### Author Response · Authors · 2022-11-07
> **Reply to reviewer sBZy**
>
> Dear reviewer,
> Thank you for the valuable comments. We would like to work our best to address your concerns answer your questions.
>
> - Regarding “CLE takes O(log n) is typo”
>
> Thanks for detecting this typo. We will carefully proofread the content in a future revision.
>
> - Regarding  “large-scale experiments”
>
> In Figure 3 we tested all the baselines for the graph with more than 1000 vertices and 10^6 edges. This is a relatively large task for arborescence learning applications.
>
> - Regarding “more comparison to Lovasz methods”.
>
> We will include more comparisons with the existing baselines. The main difference is that the classic Lovasz needs $n^3$ processors to process graphs with $n$ vertices.

---

### Official Review · Reviewer_5U91 · 2022-10-29

**Confidence:** 2
**Correctness:** 3
**Technical Novelty And Significance:** 3
**Empirical Novelty And Significance:** 3
**Recommendation:** 3

**Clarity, Quality, Novelty And Reproducibility:**

Clarity:  The paper has serious presentation issues:  while there are some grammar/syntax issues that impede understanding, the greatest issue is the lack of a clear presentation of ideas.  I found it difficult to even extract a clear problem statement/setup (and only then did it make sense as I have seen similar setups before).  Without major revisions, I cannot see voting to accept this work.

Quality:  It seems like neural networks are just randomly inserted here.  To be more specific, while the proposed improvements apply to weights determined by a neural network, I don't understand why neural networks are necessary here.  The primary result could be stated more generically, at least as far as I can tell, and the application of this result is then in dependency parsing, etc. where neural networks are used.

Novelty:  While building heavily on existing methods, the suggested improvements do seem to be novel -- though it isn't clear to me that this special case is not addressed elsewhere in other, more theoretical, works.

Reproducibility:  I found the description of everything sufficiently unclear.  I'm not confident that I could reproduce the experimental results.

**Strength And Weaknesses:**

Strengths:  The paper purports a practical improvement that makes the minimum weight arboresence solvable at larger scales, which may be of sufficient interest to certain ML subcommunities.

Weaknesses:  The exposition and presentation lack considerably when compared against typically accepted papers at ICLR.

**Summary Of The Paper:**

The authors propose an improvement to an existing algorithm for the minimum weight arboresence problem in a special class of directed graphs.  The proposed improvements are verified experimentally on both real and synthetic problems.

**Summary Of The Review:**

While the paper seems interesting overall and appears to describe a practical improvement for a class of problems, the presentation makes it difficult to read/verify its central claims.

---

### Decision · Program_Chairs · 2023-01-20

**Decision:**

Reject

**Justification For Why Not Higher Score:**

The paper is too poorly written to be accepted in its current form. The theoretical results do not give any improvements over the Lovasz methods and the empirical results are not very convincing either. The paper could be improved with more extensive experiments.

**Justification For Why Not Lower Score:**

N/A

**Metareview: Summary, Strengths And Weaknesses:**

This paper introduces a new algorithm, called the LAVA method, for computing the minimum weight arboresence problem. It is known that this problem can be reduced to the Multi-Pair Shortest Path problem. There are two standard algorithms, the Chu-Liu-Edmonds algorithm and the Lovasz methods which require $O(n)$ and $O(\log n)$ contractions respectively. The key observation is that in many machine learning applications we have complete weighted graphs, because the model predicts edge weights for all pairs of vertices. In this restricted case, they show how to speed up the algorithms. They introduce other tricks for large-scale applications. However their algorithm still requires $O(\log n)$ contractions. All of the reviewers found the paper to be poorly written and organized. The empirical results are also not very convincing.

**Summary Of Ac-Reviewer Meeting:**

N/A